# On the Sample Complexity of Subspace Learning

**Alessandro Rudi**
Robotics Brain and Cognitive Science
Istituto Italiano di Tecnologia
alessandro.rudi@iit.it

**Guillermo D. Canas**
Massachussetss Institute of Technology
guilledc@mit.edu

**Lorenzo Rosasco**
Universita' degli Studi di Genova, LCSL,
Massachusetts Institute of Technology & Istituto Italiano di Tecnologia
lrosasco@mit.edu

## Abstract

A large number of algorithms in machine learning, from principal component analysis (PCA), and its non-linear (kernel) extensions, to more recent spectral embedding and support estimation methods, rely on estimating a linear subspace from samples. In this paper we introduce a general formulation of this problem and derive novel learning error estimates. Our results rely on natural assumptions on the spectral properties of the covariance operator associated to the data distribution, and hold for a wide class of metrics between subspaces. As special cases, we discuss sharp error estimates for the reconstruction properties of PCA and spectral support estimation. Key to our analysis is an operator theoretic approach that has broad applicability to spectral learning methods.

## 1 Introduction

The subspace learning problem is that of finding the smallest linear space supporting data drawn from an unknown distribution. It is a classical problem in machine learning and statistics, with several established algorithms addressing it, most notably PCA and kernel PCA [12, 18]. It is also at the core of a number of spectral methods for data analysis, including spectral embedding methods, from classical multidimensional scaling (MDS) [7, 26], to more recent manifold embedding methods [22, 16, 2], and spectral methods for support estimation [9]. Therefore knowledge of the speed of convergence of the subspace learning problem, with respect to the sample size, and the algorithms' parameters, is of considerable practical importance.

Given a measure $\rho$ from which independent samples are drawn, we aim to estimate the smallest subspace $S_\rho$ that contains the support of $\rho$. In some cases, the support may lie on, or close to, a subspace of lower dimension than the embedding space, and it may be of interest to learn such a subspace $S_\rho$ in order to replace the original samples by their local encoding with respect to $S_\rho$.

While traditional methods, such as PCA and MDS, perform such subspace estimation in the data's original space, other, more recent manifold learning methods, such as isomap [22], Hessian eigenmaps [10], maximum-variance unfolding [24, 25, 21], locally-linear embedding [16, 17], and Laplacian eigenmaps [2] (but also kernel PCA [18]), begin by embedding the data in a *feature space*, in which subspace estimation is carried out. Indeed, as pointed out in [11, 4, 3], the algorithms in this family have a common structure. They embed the data in a suitable Hilbert space $\mathcal{H}$, and compute a linear subspace that best approximates the embedded data. The local coordinates in this subspace then become the new representation space. Similar spectral techniques may also be used to estimate the support of the data itself, as discussed in [9].

While the subspace estimates are derived from the available samples only, or their embedding, the learning problem is concerned with the quality of the computed subspace as an estimate of $S_\rho$ (the true span of the support of $\rho$). In particular, it may be of interest to understand the quality of these estimates, as a function of the algorithm's parameters (typically the dimensionality of the estimated subspace).

We begin by defining the subspace learning problem (Sec. 2), in a sufficiently general way to encompass a number of well-known problems as special cases (Sec. 4). Our main technical contribution is a general learning rate for the subspace learning problem, which is then particularized to common instances of this problem (Sec. 3). Our proofs use novel tools from linear operator theory to obtain learning rates for the subspace learning problem which are significantly sharper than existing ones, under typical assumptions, but also cover a wider range of performance metrics. A full sketch of the main proofs is given in Section 7, including a brief description of some of the novel tools developed. We conclude with experimental evidence, and discussion (Sec. 5 and 6).

## 2 Problem definition and notation

Given a measure $\rho$ with support $M$ in the unit ball of a separable Hilbert space $\mathcal{H}$, we consider in this work the problem of estimating, from $n$ i.i.d. samples $X_n = \{x_i\}_{1 \leq i \leq n}$, the smallest linear subspace $S_\rho := \overline{\text{span}(M)}$ that contains $M$.

The quality of an estimate $\hat{S}$ of $S_\rho$, for a given metric (or error criterion) $d$, is characterized in terms of probabilistic bounds of the form

$$\mathbb{P}\left[ d(S_\rho, \hat{S}) \leq \varepsilon(\delta, n, \rho) \right] \geq 1 - \delta, \quad 0 < \delta \leq 1. \tag{1}$$

for some function $\varepsilon$ of the problem's parameters. We derive in the sequel high probability bounds of the above form.

In the remainder the metric projection operator onto a subspace $S$ is denoted by $P_S$, where $P_S^2 = P_S^* = P_S$ (every $P$ is idempotent and self-adjoint). We denote by $\|\cdot\|_\mathcal{H}$ the norm induced by the dot product $< \cdot, \cdot >_\mathcal{H}$ in $\mathcal{H}$, and by $\|A\|_p := \sqrt[p]{\text{Tr}(|A|^p)}$ the $p$-Schatten, or $p$-class norm of a linear bounded operator $A$ [15, p. 84].

### 2.1 Subspace estimates

Letting $C := \mathbb{E}_{x \sim \rho} x \otimes x$ be the (uncentered) covariance operator associated to $\rho$, it is easy to show that $S_\rho = \overline{\text{Ran } C}$. Similarly, given the empirical covariance $C_n := \frac{1}{n} \sum_{i=1}^n x \otimes x$, we define the *empirical subspace estimate*

$$\hat{S}_n := \text{span}(X_n) = \text{Ran } C_n$$

(note that the closure is not needed in this case because $\hat{S}_n$ is finite-dimensional). We also define the *k-truncated (kernel) PCA subspace estimate* $\hat{S}_n^k := \text{Ran } C_n^k$, where $C_n^k$ is obtained from $C_n$ by keeping only its $k$ top eigenvalues. Note that, since the PCA estimate $\hat{S}_n^k$ is spanned by the top $k$ eigenvectors of $C_n$, then clearly $\hat{S}_n^k \subseteq \hat{S}_n^{k'}$ for $k < k'$, and therefore $\{\hat{S}_n^k\}_{k=1}^n$ is a nested family of subspaces (all of which are contained in $S_\rho$).

As discussed in Section 4.1, since kernel-PCA reduces to regular PCA in a feature space [18] (and can be computed with knowledge of the kernel alone), the following discussion applies equally to kernel-PCA estimates, with the understanding that, in that case, $S_\rho$ is the span of the support of $\rho$ *in the feature space*.

### 2.2 Performance criteria

In order for a bound of the form of Equation (1) to be meaningful, a choice of performance criteria $d$ must be made. We define the distance

$$d_{\alpha,p}(U, V) := \|(P_U - P_V)C^\alpha\|_p \tag{2}$$

between subspaces $U, V$, which is a metric over the space of subspaces contained in $S_\rho$, for $0 \leq \alpha \leq \frac{1}{2}$ and $1 \leq p \leq \infty$. Note that $d_{\alpha,p}$ depends on $\rho$ through $C$ but, in the interest of clarity,

this dependence is omitted in the notation. While of interest in its own right, it is also possible to express important performance criteria as particular cases of $d_{\alpha,p}$. In particular, the so-called **reconstruction error** [13]:

$$d_R(S_\rho, \hat{S}) := \mathbb{E}_{x \sim \rho} \| P_{S_\rho}(x) - P_{\hat{S}}(x) \|_{\mathcal{H}}^2$$

is $d_R(S_\rho, \cdot) = d_{1/2,2}(S_\rho, \cdot)^2$.

Note that $d_R$ is a natural criterion because a $k$-truncated PCA estimate minimizes a suitable error $d_R$ over all subspaces of dimension $k$. Clearly, $d_R(S_\rho, \hat{S})$ vanishes whenever $\hat{S}$ contains $S_\rho$ and, because the family $\{\hat{S}_n^k\}_{k=1}^n$ of PCA estimates is nested, then $d_R(S_\rho, \hat{S}_n^k)$ is non-increasing with $k$.

As shown in [13], a number of unsupervised learning algorithms, including (kernel) PCA, k-means, k-flats, sparse coding, and non-negative matrix factorization, can be written as a minimization of $d_R$ over an algorithm-specific class of sets (e.g. over the set of linear subspaces of a fixed dimension in the case of PCA).

## 3 Summary of results

Our main technical contribution is a bound of the form of Eq. (1), for the $k$-truncated PCA estimate $\hat{S}_n^k$ (with the empirical estimate $\hat{S}_n := \hat{S}_n^n$ being a particular case), whose proof is postponed to Sec. 7. We begin by bounding the distance $d_{\alpha,p}$ between $S_\rho$ and the $k$-truncated PCA estimate $\hat{S}_n^k$, given a known covariance $C$.

**Theorem 3.1.** *Let $\{x_i\}_{1 \leq i \leq n}$ be drawn i.i.d. according to a probability measure $\rho$ supported on the unit ball of a separable Hilbert space $\mathcal{H}$, with covariance $C$. Assuming $n > 3$, $0 < \delta < 1$, $0 \leq \alpha \leq \frac{1}{2}$, $1 \leq p \leq \infty$, the following holds for each $k \in \{1, \ldots, n\}$:*

$$\mathbb{P}\left[ d_{\alpha,p}(S_\rho, \hat{S}_n^k) \leq 3t^\alpha \left\| C^\alpha (C + tI)^{-\alpha} \right\|_p \right] \geq 1 - \delta \tag{3}$$

*where $t = \max\{\sigma_k, \frac{9}{n} \log \frac{n}{\delta}\}$, and $\sigma_k$ is the $k$-th top eigenvalue of $C$.*

We say that $C$ has *eigenvalue decay rate of order $r$* if there are constants $q, Q > 0$ such that $qj^{-r} \leq \sigma_j \leq Qj^{-r}$, where $\sigma_j$ are the (decreasingly ordered) eigenvalues of $C$, and $r > 1$. From Equation (2) it is clear that, in order for the subspace learning problem to be well-defined, it must be $\|C^\alpha\|_p < \infty$, or alternatively: $\alpha p > 1/r$. Note that this condition is always met for $p = \infty$, and also holds in the reconstruction error case ($\alpha = 1/2, p = 2$), for any decay rate $r > 1$.

Knowledge of an eigenvalue decay rate can be incorporated into Theorem 3.1 to obtain explicit learning rates, as follows.

**Theorem 3.2** (Polynomial eigenvalue decay). *Let $C$ have eigenvalue decay rate of order $r$. Under the assumptions of Theorem 3.1, it is, with probability $1 - \delta$*

$$d_{\alpha,p}(S_\rho, \hat{S}_n^k) \leq \begin{cases} Q' k^{-r\alpha + \frac{1}{p}} & \text{if } k < k_n^* & \text{(polynomial decay)} \\ Q' k_n^{*-r\alpha + \frac{1}{p}} & \text{if } k \geq k_n^* & \text{(plateau)} \end{cases} \tag{4}$$

*where it is $k_n^* = \left( \frac{qn}{9 \log(n/\delta)} \right)^{1/r}$, and $Q' = 3 \left( Q^{1/r} \Gamma(\alpha p - 1/r) \Gamma(1 + 1/r) / \Gamma(1/r) \right)^{1/p}$.*

The above theorem guarantees a drop in $d_{\alpha,p}$ with increasing $k$, at a rate of $k^{-r\alpha + 1/p}$, up to $k = k_n^*$, after which the bound remains constant. The estimated plateau threshold $k^*$ is thus the value of truncation past which the upper bound does not improve. Note that, as described in Section 5, this performance drop and plateau behavior is observed in practice.

The proofs of Theorems 3.1 and 3.2 rely on recent non-commutative Bernstein-type inequalities on operators [5, 23], and a novel analytical decomposition. Note that classical Bernstein inequalities in Hilbert spaces (e.g. [14]) could also be used instead of [23]. However, while this approach would simplify the analysis, it produces looser bounds, as described in Section 7.

If we consider an algorithm that produces, for each set of $n$ samples, an estimate $\hat{S}_n^k$ with $k \geq k_n^*$ then, by plugging the definition of $k_n^*$ into Eq. 4, we obtain an upper bound on $d_{\alpha,p}$ as a function of $n$.

**Corollary 3.3.** *Let $C$ have eigenvalue decay rate of order $r$, and $Q'$, $k_n^*$ be as in Theorem 3.2. Let $\hat{S}_n^*$ be a truncated subspace estimate $\hat{S}_n^k$ with $k \geq k_n^*$. It is, with probability $1 - \delta$,*

$$d_{\alpha,p}(S_\rho, \hat{S}_n^*) \leq Q' \left( \frac{9 \left( \log n - \log \delta \right)}{qn} \right)^{\alpha - \frac{1}{rp}}$$

**Remark 3.4.** *Note that, by setting $k = n$, the above corollary also provides guarantees on the rate of convergence of the empirical estimate $S_n = span(X_n)$ to $S_\rho$, of order*

$$d_{\alpha,p}(S_\rho, S_n) = O \left( \left( \frac{\log n - \log \delta}{n} \right)^{\alpha - \frac{1}{rp}} \right).$$

Corollary 4.1 and remark 3.4 are valid for all $n$ such that $k_n^* \leq n$ (or equivalently such that $n^{r-1}(\log n - \log \delta) \geq q/9$). Note that, because $\rho$ is supported on the unit ball, its covariance has eigenvalues no greater than one, and therefore it must be $q < 1$. It thus suffices to require that $n > 3$ to ensure the condition $k_n^* \leq n$ to hold.

## 4  Applications of subspace learning

We describe next some of the main uses of subspace learning in the literature.

### 4.1  Kernel PCA and embedding methods

One of the main applications of subspace learning is in reducing the dimensionality of the input. In particular, one may find nested subspaces of dimension $1 \leq k \leq n$ that minimize the distances from the original to the projected samples. This procedure is known as the Karhunen-Loève, PCA, or Hotelling transform [12], and has been generalized to Reproducing-Kernel Hilbert Spaces (RKHS) [18].

In particular, the above procedure amounts to computing an eigen-decomposition of the empirical covariance (Sec. 2.1):

$$C_n = \sum_{i=1}^{n} \sigma_i u_i \otimes u_i,$$

where the $k$-th subspace estimate is $\hat{S}_n^k := \operatorname{Ran} C_n^k = \operatorname{span}\{u_i : 1 \leq i \leq k\}$. Note that, in the general case of kernel PCA, we assume the samples $\{x_i\}_{1 \leq i \leq n}$ to be in some RKHS $\mathcal{H}$, which are obtained from the observed variables $(z_1, \ldots, z_n) \in Z^n$, for some space $Z$, through an embedding $x_i := \phi(z_i)$. Typically, due to the very high dimensionality of $\mathcal{H}$, we may only have indirect information about $\phi$ in the form a kernel function $K : Z \times Z \to \mathbb{R}$: a symmetric, positive definite function satisfying $K(z, w) = \langle \phi(z), \phi(w) \rangle_{\mathcal{H}}$ [20] (for technical reasons, we also assume $K$ to be continuous). Note that every such $K$ has a unique associated RKHS, and viceversa [20, p. 120–121], whereas, given $K$, the embedding $\phi$ is only unique up to an inner product-preserving transformation.

Given a point $z \in Z$, we can make use of $K$ to compute the coordinates of the projection of its embedding $\phi(z)$ onto $\hat{S}_n^k \subseteq \mathcal{H}$ by means of a simple $k$-truncated eigen-decomposition of $K_n$.

It is easy to see that the $k$-truncated kernel PCA subspace $\hat{S}_n^k$ minimizes the empirical reconstruction error $d_R(\hat{S}_n, \hat{S})$, among all subspaces $\hat{S}$ of dimension $k$. Indeed, it is

$$\begin{aligned}
d_R(\hat{S}_n, \hat{S}) &= \mathbb{E}_{x \sim \hat{\rho}} \|x - P_{\hat{S}}(x)\|_{\mathcal{H}}^2 = \mathbb{E}_{x \sim \hat{\rho}} \left\langle (I - P_{\hat{S}})x, (I - P_{\hat{S}})x \right\rangle_{\mathcal{H}} \\
&= \mathbb{E}_{x \sim \hat{\rho}} \left\langle I - P_{\hat{S}}, x \otimes x \right\rangle_{HS} = \left\langle I - P_{\hat{S}}, C_n \right\rangle_{HS},
\end{aligned} \tag{5}$$

where $\langle \cdot, \cdot \rangle_{HS}$ is the Hilbert-Schmidt inner product, from which it is easy to see that the $k$-dimensional subspace minimizing Equation 5 (alternatively maximizing $< P_{\hat{S}}, C_n >$) is spanned by the $k$-top eigenvectors of $C_n$.

Since we are interested in the expected $d_R(S_\rho, \hat{S}_n^k)$ (rather than the empirical $d_R(\hat{S}_n, \hat{S})$) error of the kernel PCA estimate, we may obtain a learning rate for Equation 5 by particularizing Theorem 3.2

to the reconstruction error, for all $k$ (Theorem 3.2), and for $k \geq k^*$ with a suitable choice of $k^*$ (Corollary 4.1). In particular, recalling that $d_R(S_\rho, \cdot) = d_{\alpha,p}(S_\rho, \cdot)^2$ with $\alpha = 1/2$ and $p = 2$, and choosing a value of $k \geq k_n^*$ that minimizes the bound of Theorem 3.2, we obtain the following result.

**Corollary 4.1** (Performance of PCA / Reconstruction error). *Let $C$ have eigenvalue decay rate of order $r$, and $\hat{S}_n^*$ be as in Corollary 3.3. Then it holds, with probability $1 - \delta$,*

$$d_R(S_\rho, \hat{S}_n^*) = O\left( \left( \frac{\log n - \log \delta}{n} \right)^{1-1/r} \right)$$

*where the dependence on $\delta$ is hidden in the Landau symbol.*

## 4.2 Support estimation

The problem of support estimation consists in recovering the support $M$ of a distribution $\rho$ on a metric space $Z$ from identical and independent samples $Z_n = (z_i)_{1 \leq i \leq n}$. We briefly recall a recently proposed approach to support estimation based on subspace learning [9], and discuss how our results specialize to this setting, producing a qualitative improvement to theirs.

Given a suitable reproducing kernel $K$ on $Z$ (with associated feature map $\phi$), the support $M$ can be characterized in terms of the subspace $S_\rho = \overline{\text{span}\, \phi(M)} \subseteq \mathcal{H}$ [9]. More precisely, letting $d_V(x) = \|x - P_V x\|_{\mathcal{H}}$ be the point-subspace distance to a subspace $V$, it can be shown (see [9]) that, if the kernel *separates* [1] $M$, then it is

$$M = \{z \in Z \mid d_{S_\rho}(\phi(z)) = 0\}.$$

This suggests an empirical estimate $\hat{M} = \{z \in Z \mid d_{\hat{S}}(\phi(z)) \leq \tau\}$ of $M$, where $\hat{S} = \overline{\text{span}\, \phi(Z_n)}$, and $\tau > 0$. With this choice, almost sure convergence $\lim_{n\to\infty} d_H(M, \hat{M}) = 0$ in the Hausdorff distance [1] is related to the convergence of $\hat{S}$ to $S_\rho$ [9]. More precisely, if the eigenfunctions of the covariance operator $C = \mathbb{E}_{z\sim\rho}[\phi(z) \otimes \phi(z)]$ are uniformly bounded, then it suffices for Hausdorff convergence to bound from above $d_{\frac{r-1}{2r},\infty}$ (where $r > 1$ is the eigenvalue decay rate of $C$). The following results specializes Corollary 3.3 to this setting.

**Corollary 4.2** (Performance of set learning). *If $0 \leq \alpha \leq \frac{1}{2}$, then it holds, with probability $1 - \delta$,*

$$d_{\alpha,\infty}(S_\rho, \hat{S}_n^*) = O\left( \left( \frac{\log n - \log \delta}{n} \right)^{\alpha} \right)$$

*where the constant in the Landau symbol depends on $\delta$.*

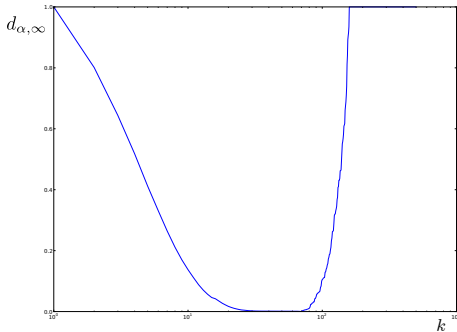

Figure 1: The figure shows the experimental behavior of the distance $d_{\alpha,\infty}(\hat{S}^k, S_\rho)$ between the empirical and the actual support subspaces, with respect to the regularization parameter. The setting is the one of section 5. Here the actual subspace is analytically computed, while the empirical one is computed on a dataset with $n = 1000$ and 32bit floating point precision. Note the numerical instability as $k$ tends to 1000.

Letting $\alpha = \frac{r-1}{2r}$ above yields a high probability bound of order $O\left(n^{-\frac{r-1}{2r}}\right)$ (up to logarithmic factors), which is considerably sharper than the bound $O\left(n^{-\frac{r-1}{2(3r-1)}}\right)$ found in [8] (Theorem 7).

Note that these are upper bounds for the best possible choice of $k$ (which minimizes the bound). While the optima of both bounds vanish with $n \to \infty$, their behavior is qualitatively different. In particular, the bound of [8] is U-shaped, and diverges for $k = n$, while ours is L-shaped (no trade-off), and thus also convergent for $k = n$. Therefore, when compared with [8], our results suggest that no regularization is required from a statistical point of view though, as clarified in the following remark, it may be needed for purposes of numerical stability.

**Remark 4.3.** *While, as proven in Corollary 4.2, regularization is not needed from a statistical perspective, it can play a role in ensuring numerical stability in practice. Indeed, in order to find $\hat{M}$, we compute $d_{\hat{S}}(\phi(z))$ with $z \in Z$. Using the reproducing property of $K$, it can be shown that, for $z \in Z$, it is $d_{\hat{S}^k}(\phi(z)) = K(z, z) - \left\langle t_z, (\hat{K}_n^k)^\dagger t_z \right\rangle$ where $(t_z)_i = K(z, z_i)$, $\hat{K}_n$ is the Gram matrix $(\hat{K}_n)_{ij} = K(z_i, z_j)$, $\hat{K}_n^k$ is the rank-$k$ approximation of $\hat{K}_n$, and $(\hat{K}_n^k)^\dagger$ is the pseudo-inverse of $\hat{K}_n^k$. The computation of $\hat{M}$ therefore requires a matrix inversion, which is prone to instability for high condition numbers. Figure 1 shows the behavior of the error that results from replacing $\hat{S}$ by its $k$-truncated approximation $\hat{S}^k$. For large values of $k$, the small eigenvalues of $\hat{S}$ are used in the inversion, leading to numerical instability.*

# 5  Experiments

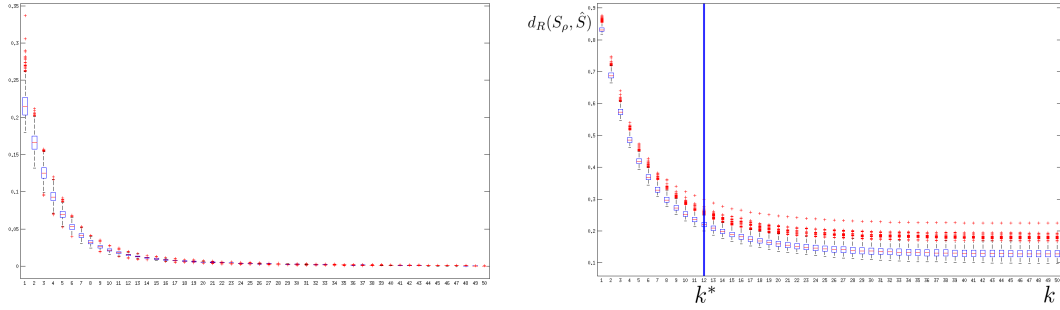

Figure 2: The spectrum of the empirical covariance (left), and the expected distance from a random sample to the empirical $k$-truncated kernel-PCA subspace estimate (right), as a function of $k$ ($n = 1000$, 1000 trials shown in a boxplot). Our predicted plateau threshold $k_n^*$ (Theorem 3.2) is a good estimate of the value $k$ past which the distance stabilizes.

In order to validate our analysis empirically, we consider the following experiment. Let $\rho$ be a uniform one-dimensional distribution in the unit interval. We embed $\rho$ into a reproducing-kernel Hilbert space $\mathcal{H}$ using the exponential of the $\ell_1$ distance ($k(u, v) = \exp\{-\|u - v\|_1\}$) as kernel. Given $n$ samples drawn from $\rho$, we compute its empirical covariance in $\mathcal{H}$ (whose spectrum is plotted in Figure 2 (left)), and truncate its eigen-decomposition to obtain a subspace estimate $\hat{S}_n^k$, as described in Section 2.1.

Figure 2 (right) is a box plot of reconstruction error $d_R(S_\rho, \hat{S}_n^k)$ associated with the $k$-truncated kernel-PCA estimate $\hat{S}_n^k$ (the expected distance in $\mathcal{H}$ of samples to $\hat{S}_n^k$), with $n = 1000$ and varying $k$. While $d_R$ is computed analytically in this example, and $S_\rho$ is fixed, the estimate $\hat{S}_n^k$ is a random variable, and hence the variability in the graph. Notice from the figure that, as pointed out in [6] and discussed in Section 6, the reconstruction error $d_R(S_\rho, \hat{S}_n^k)$ is always a non-increasing function of $k$, due to the fact that the kernel-PCA estimates are nested: $\hat{S}_n^k \subset \hat{S}_n^{k'}$ for $k < k'$ (see Section 2.1). The graph is highly concentrated around a curve with a steep intial drop, until reaching some sufficiently high $k$, past which the reconstruction (pseudo) distance becomes stable, and does not vanish. In our experiments, this behavior is typical for the reconstruction distance and high-dimensional problems.

Due to the simple form of this example, we are able to compute analytically the spectrum of the true covariance $C$. In this case, the eigenvalues of $C$ decay as $2\gamma/((k\pi)^2 + \gamma^2)$, with $k \in \mathbb{N}$, and therefore they have a polynomial decay rate $r = 2$ (see Section 3). Given the known spectrum decay rate, we can estimate the plateau threshold $k = k_n^*$ in the bound of Theorem 3.2, which can be seen

to be a good approximation of the observed start of a plateau in $d_R(S_\rho, \hat{S}_n^k)$ (Figure 2, right). Notice that our bound for this case (Corollary 4.1) similarly predicts a steep performance drop until the threshold $k = k_n^*$ (indicated in the figure by the vertical blue line), and a plateau afterwards.

## 6   Discussion

Figure 3 shows a comparison of our learning rates with existing rates in the literature [6, 19]. The plot shows the polynomial decay rate $c$ of the high probability bound $d_R(S_\rho, \hat{S}_n^k) = O(n^{-c})$, as a function of the eigenvalue decay rate $r$ of the covariance $C$, computed at the best value $k_n^*$ (which minimizes the bound).

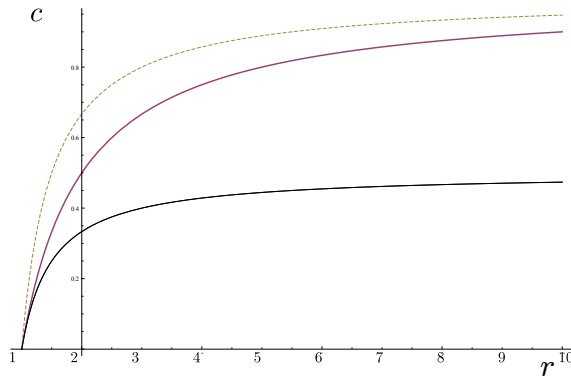

Figure 3:   Known upper bounds for the polynomial decay rate $c$ (for the best choice of $k$), for the expected distance from a random sample to the empirical $k$-truncated kernel-PCA estimate, as a function of the covariance eigenvalue decay rate (higher is better). Our bound (purple line), consistently outperforms previous ones [19] (black line). The top (dashed) line [6], has significantly stronger assumptions, and is only included for completeness.

The learning rate exponent $c$, under a polynomial eigenvalue decay assumption of the data covariance $C$, is $c = \frac{s(r-1)}{r-s+sr}$ for [6] and $c = \frac{r-1}{2r-1}$ for [19], where $s$ is related to the fourth moment. Note that, among the two (purple and black) that operate under the same assumptions, our bound (purple line) is the best by a wide margin. The top, best performing, dashed line [6] is obtained for the best possible fourth-order moment constraint $s = 2r$, and is therefore not a fair comparison. However, it is worth noting that our bounds perform almost as well as the most restrictive one, even when we do not include any fourth-order moment constraints.

**Choice of truncation parameter** $k$. Since, as pointed out in Section 2.1, the subspace estimates $\hat{S}_n^k$ are nested for increasing $k$ (i.e. $\hat{S}_n^k \subseteq \hat{S}_n^{k'}$ for $k < k'$), the distance $d_{\alpha,p}(S_\rho, \hat{S}_n^k)$, and in particular the reconstruction error $d_R(S_\rho, \hat{S}_n^k)$, is a non-increasing function of $k$. As has been previously discussed [6], this suggests that there is no tradeoff in the choice of $k$. Indeed, the fact that the estimates $\hat{S}_n^k$ become increasing close to $S_\rho$ as $k$ increases indicates that, when minimizing $d_{\alpha,p}(S_\rho, \hat{S}_n^k)$, the best choice is the highest: $k = n$.

Interestingly, however, both in practice (Section 5), and in theory (Section 3), we observe that a typical behavior for the subspace learning problem in high dimensions (e.g. kernel PCA) is that there is a certain value of $k = k_n^*$, past which performance plateaus. For problems such as spectral embedding methods [22, 10, 25], in which a degree of dimensionality reduction is desirable, producing an estimate $\hat{S}_n^k$ where $k$ is close to the plateau threshold may be a natural parameter choice: it leads to an estimate of the lowest dimension ($k = k_n^*$), whose distance to the true $S_\rho$ is almost as low as the best-performing one ($k = n$).

# 7 Sketch of the proofs

Due to the novelty of the the techniques employed, and in order to clarify how they may be used in other contexts, we provide here a proof of our main theoretical result, Theorem 3.1, with some details omitted in the interest of conciseness.

For each $\lambda > 0$, we denote by $r^\lambda(x) := \mathbf{1}\{x > \lambda\}$ the step function with a cut-off at $\lambda$. Given an empirical covariance operator $C_n$, we will consider the truncated version $r^\lambda(C_n)$ where, in this notation, $r^\lambda$ is applied to the eigenvalues of $C_n$, that is, $r^\lambda(C_n)$ has the same eigen-structure as $C_n$, but its eigenvalues that are less or equal to $\lambda$ are clamped to zero.

In order to prove the bound of Equation (3), we begin by proving a more general upper bound of $d_{\alpha,p}(S_\rho, \hat{S}_n^k)$, which is split into a random ($\mathcal{A}$), and a deterministic part ($\mathcal{B}, \mathcal{C}$). The bound holds for all values of a free parameter $t > 0$, which is then constrained and optimized in order to find the (close to) tightest version of the bound.

**Lemma 7.1.** *Let $t > 0$, $0 \le \alpha \le 1/2$, and $\lambda = \sigma_k(C)$ be the k-th top eigenvalue of $C$, it is,*

$$d_{\alpha,p}(S_\rho, \hat{S}_n^k) \le \underbrace{\|(C + tI)^{\frac{1}{2}}(C_n + tI)^{-\frac{1}{2}}\|_\infty^{2\alpha}}_{\mathcal{A}} \cdot \underbrace{\{3/2(\lambda + t)\}^\alpha}_{\mathcal{B}} \cdot \underbrace{\|C^\alpha(C + tI)^{-\alpha}\|_p}_{\mathcal{C}} \quad (6)$$

Note that the right-hand side of Equation (6) is the product of three terms, the left of which ($\mathcal{A}$) involves the empirical covariance operator $C_n$, which is a random variable, and the right two ($\mathcal{B}, \mathcal{C}$) are entirely deterministic. While the term $\mathcal{B}$ has already been reduced to the known quantities $t, \alpha, \lambda$, the remaining terms are bound next. We bound the random term $\mathcal{A}$ in the next Lemma, whose proof makes use of recent concentration results [23].

**Lemma 7.2** (Term $\mathcal{A}$). *Let $0 \le \alpha \le 1/2$, for each $\frac{9}{n}\log\frac{n}{\delta} \le t \le \|C\|_\infty$, with probability $1 - \delta$ it is*

$$(2/3)^\alpha \le \|(C + tI)^{\frac{1}{2}}(C_n + tI)^{-\frac{1}{2}}\|_\infty^{2\alpha} \le 2^\alpha$$

**Lemma 7.3** (Term $\mathcal{C}$). *Let $C$ be a symmetric, bounded, positive semidefinite linear operator on $\mathcal{H}$. If $\sigma_k(C) \le f(k)$ for $k \in \mathbb{N}$, where $f$ is a decreasing function then, for all $t > 0$ and $\alpha \ge 0$, it holds*

$$\left\|C^\alpha(C + tI)^{-\alpha}\right\|_p \le \inf_{0 \le u \le 1} g_{u\alpha} t^{-u\alpha} \quad (7)$$

*where $g_{u\alpha} = \left(f(1)^{u\alpha p} + \int_1^\infty f(x)^{u\alpha p} dx\right)^{1/p}$. Furthermore, if $f(k) = gk^{-1/\gamma}$, with $0 < \gamma < 1$ and $\alpha p > \gamma$, then it holds*

$$\left\|C^\alpha(C + tI)^{-\alpha}\right\|_p \le Qt^{-\gamma/p} \quad (8)$$

*where $Q = (g^\gamma \Gamma(\alpha p - \gamma)\Gamma(1 + \gamma)/\Gamma(\gamma))^{1/p}$.*

The combination of Lemmas 7.1 and 7.2 leads to the main theorem 3.1, which is a probabilistic bound, holding for every $k \in \{1, \ldots, n\}$, with a deterministic term $\|C^\alpha(C + tI)^{-\alpha}\|_p$ that depends on knowledge of the covariance $C$. In cases in which some knowledge of the decay rate of $C$ is available, Lemma 7.3 can be applied to obtain Theorem 3.2 and Corollary 3.3. Finally, Corollary 4.1 is simply a particular case for the reconstruction error $d_R(S_\rho, \cdot) = d_{\alpha,p}(S_\rho, \cdot)^2$, with $\alpha = 1/2, p = 2$.

As noted in Section 3, looser bounds would be obtained if classical Bernstein inequalities in Hilbert spaces [14] were used instead. In particular, Lemma 7.2 would result in a range for $t$ of $qn^{-r/(r+1)} \le t \le \|C\|_\infty$, implying $k^* = O(n^{1/(r+1)})$ rather than $O(n^{1/r})$, and thus Theorem 3.2 would become (for $k \ge k^*$) $d_{\alpha,p}(S_\rho, S_n^k) = O(n^{-\alpha r/(r+1)+1/(p(r+1))})$ (compared with the sharper $O(n^{-\alpha+1/rp})$ of Theorem 3.2). For instance, for $p = 2$, $\alpha = 1/2$, and a decay rate $r = 2$ (as in the example of Section 5), it would be: $d_{1/2,2}(S_\rho, S_n) = O(n^{-1/4})$ using Theorem 3.2, and $d_{1/2,2}(S_\rho, S_n) = O(n^{-1/6})$ using classical Bernstein inequalities.

**Acknowledgments** L. R. acknowledges the financial support of the Italian Ministry of Education, University and Research FIRB project RBFR12M3AC.

## Footnotes

[1] A kernel is said to separate $M$ if its associated feature map $\phi$ satisfies $\phi^{-1}(\overline{\text{span}\, \phi(M)}) = M$ (e.g. the Abel kernel is separating).

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
