[Reviews · NeurIPS 2013]

Submitted by Assigned_Reviewer_5

Paper Summary:
Authors deal with a fundamental problem of deriving PAC-style rates for subspace estimation using samples. The key result shows for n samples drawn from some underlying distribution, the quality of subspace estimation improves at a rate O(n^-r), where r related to the decay rate of the spectrum of the underlying distribution.

Review:
I am not familiar with the previous literature on PAC-style analysis of subspace learning, or if properties of the spectrum of the covariance was previously considered for subspace learning; so assuming that the work is novel, I believe authors have done a good job in relating these concepts.

I do have a few suggestions that the authors should consider adding to the current text:

Although authors have focused on the theoretical aspects of subspace learning, it would be nice to see how well the condition of ‘polynomial decay’ holds on real world data. This would help with the significance of this work to the larger machine learning audience.

Going a step further, it would be very instructive to see what the rates look like when the covariance C is unknown. That is, is it possible to give the error rate in terms of _observed_ covariance C_n and an empirical decay rate ‘r_n’ (instead of the true covariance C and decay rate r)? This would certainly be helpful when dealing with real data where one only has access to a sample from some underlying measure. I would encourage the authors to at least add a brief discussion on this.

Theorem 3.1 in its current form seems to be for a fixed value of k, again for practical purposes, k is often chosen based on the samples, so perhaps a uniform bound on k would be more useful?

Why suppress the dependence on \delta in Corollary 4.1 and 4.2? I believe it would be more instructive to see the explicit dependence O( log(n/delta)/n ^ …). I would suggest to move the generic rate to the introduction, and give explicit rate with all the constants in the corollaries.

I have to admit that given the short review period, I did not have a chance to check the proofs.


Quality:
This is a theoretical paper, with main proof using concepts from operator theory. Given the short review period, I did not have a chance to verify the proof.

Clarity:
The paper is written clearly.

Originality:
I am not familiar with the past literature, and cannot comment on the originality of this work.

Significance:
Given the ubiquity of subspace finding methods, I believe work is very significant as it provides the finite sample rates for subspace learning.
Summary: Although I didn't have chance to check the proofs, if the proofs check out, I believe this work would be a nice paper for the theoretically inclined audience. I do encourage the authors to consider the suggestions made in the review.

Submitted by Assigned_Reviewer_6

The paper derives error estimates for subspace learning under spectral assumptions on covariance operators.

The paper makes a solid impression -- though you have minor things like missing words, inner product symbols, missing axes labels etc which should be corrected.

The problem addressed by the authors is certainly of relevance. And they have a number of findings that seem useful:

- they derive convergence rates for the (truncated) PCA which guarantee with a high probability that the support of the measure is retrieved with an error of ~ n^-(1-1/r) and -r is the exponent of the decay of the eigenvalues of the associated covariance operator

- for support estimation they gain a factor of 3 in the exponent

- they state that their bound is a better predictor for the decay rate c for the expected distance from samples to the inferred sub space estimate.

- they find an estimate from which on the distance of inferred subspace to the real one does not shrink significantly

Some detailed comments:

- would be good to specify the support M in the unit ball B exactly. I guess this just means that rho(H\B) negligible

- or better what is the support M? Should that be like the smallest measurable set M such that H\M is negligible. Does such a set exist or only up to measure zero; I mean nothing big here but would be nice to have an exact definition as M is your main object of investigation.

- eq (2): I would like a bit of an discussion of the parameters alpha and p. Also is there a case beside 1/2,2 that is of relevance, and why?

- directly after eq 2: I guess you mean here closed subspaces so that the projectors are well defined;

- eq (3): here a discussion of the norm/the constant seems needed. The issue is that you don't converge to zero in your leading term but to sigma_k so the constants are not unimportant.

- Thm 3.2: didn't check the details but I'm wondering if this is for a specific k or uniform over all k; I guess the former and it would be good to point this out. Something like "given k" ...

- Cor 3.3 * -> k

Summary: A solid paper which derives error rates for subspace learning which improve over state of the art rates.

Submitted by Assigned_Reviewer_9

This paper deals with estimating the subspace on which an unknown sampling distribution is supported. I found the exact problem statement somewhat unclear. Quoting the 2nd paragraph:
<<
Given a measure rho from which independent samples are drawn, we aim to estimate the smallest subspace S_rho that contains the support of rho. In some cases, the support may lie on, or close to, a subspace of lower dimension than the embedding space, and it may be of interest to learn such a subspace S_rho in order to replace the original samples by their local encoding with respect to S_rho.
>>

What I find confusing here is that the support of rho can never be smaller than the sample support. Certainly, the sample points may lie on a strict subspace of the embedding space, but this is trivially discovered (via PCA, for example). What (lossless) dimensionality reduction is possible beyond that? Perhaps you intend to recover an "approximate" support, or an "effective" support, but this needs to be clearly stated early on.

The English is mostly fine but occasionally incorrect usage leads to comprehension problems. In particular, the authors tend to write "it is" instead of "we have that, "it is the case that" or some similar expression. [E.g., lines 084, 158, 233.]

The result are technical and I did not have time to verify their correctness, but they seem plausible. However, the motivation and intuitive explanation is sorely lacking. What is the significance of the particular distance metric you chose? What about the eigenvalue decay condition? Do such conditions hold in practice?
Summary: The paper should be rewritten with an eye to clarity, motivation, intuition, grammar.

Submitted by Meta_Reviewer_2

The paper deals with the problem of linear subspace estimation from samples, which appears in many applications such as PCA, kernel PCA and support estimation. The main contribution of the paper is that it provides a general formulation for the problem of subspace estimation and provides learning rates assuming certain decay behaviour for the eigenvalues of the covariance operator associated with the data distribution. As special cases, learning rates are provided for PCA and support estimation improving upon the existing results.

Clarity: While the paper is technically interesting, the motivation presented in Section 1 is not clear. The paper is more of a technical exercise and lacks intuitive explanation. A better approach would have been to introduce the PCA problem (as discussed in Section 4.1) in Section 2 or may be even elaborate it in Section 1 so that the reader is introduced to the operator theoretic interpretation of PCA. With this motivation, one can generalize this notion as it is done in Section 2. I think this will improve the readability of the paper. As a minor comment, there are few typos, e.g., usage of "it is", which need to be fixed.

Originality and Significance: The paper is technically interesting and is mathematically correct.

Other comments:
1. In the paragraph below Eq. 2, it is mentioned that d_{\alpha,p} is a metric for 0\le\alpha\le 1/2 and 1\le p\le\infty. However, in Proposition D.1 says that it is also a metric for 1/2\le \alpha\le 1. Its not clear why this restriction to 0\le\alpha\le 1/2? I dont see that such a restriction is needed at least at this point.

2. In the proof of Proposition D.1, it is mentioned that S_\rho=Ran C. Isn't it the closure of Ran C by Proposition D.3?

3. As mentioned above, it will be better to motivate the work through PCA and use proposition D.4 to motivate Eq. 2. While one might wonder are there any interesting values of \alpha and p other than 1/2 and 2, then you can present the application of set estimation where \alpha and p take values other than these.

4. In theorem 3.2, corollary 3.3 and remark 3.4, it is not mentioned that \alpha > 1/(rp). Without this condition, the estimator will not be consistent. This means for a fixed \alpha, p > 1/(r\alpha) which means the results do not hold for all 0\le p\le \infty as mentioned in the statements. Similar is the case with $\alpha$ where for a fixed p, the results hold only for \alpha > 1/(rp), or it should be assumed that r > max(1,1/(p\alpha))

5. Line 126: postponed to "Section 7"

6. The discussion about the non-requirement of regularizer from a statistical view point is not clear. Can you please elaborate what is making the difference compared to [11], that the proposed formulation does not require a regularizer even from statistical point of view. It is clear that some regularizer is needed for numerical stability but the lack of it is surprising from statistical sense.

7. While the proof technique is interesting and non-trivial, it appears that one can just do the entire analysis with p=2, i.e., Hilbert-Schmidt norm. Since the Schatten p-norm is a p-norm of the sequence of singular values (which can be countable in number as the operators considered are compact), it easy to check that ||A||_q\le ||A||_p for any 0 < p < q\le infty (because of the inclusion of l^p space in l^q). This means the analysis can simply by carried out using p=2 for any while bouding ||A||_p for any p\ge 2. This means classical Bernstein's inequality in Hilbert spaces can be used without needing sophisticated tools like Theorem B.1. Similarly for the case with 1\le p\le 2, one can bound ||A||_p by ||A||_1 in deriving from (14) onwards where one can use Holder and convert the analysis into bounding HS norms of certain operators.

While I do not find anything wrong the current analysis and it is indeed nice, I am just wondering whether one needs the approach used in the paper to get sharp rates or one can still get such rates using the above mentioned analysis using HS norms. If one really needs the approach used in the paper to get sharp rates, this has to be highlighted along with a possible reason showing where HS approach fails.
Summary: A general formulation for the problem of learning subspace from samples is proposed along with learning rates under certain conditions on the spectrum of the covariance operator associated with the data distribution. The paper is theoretically interesting, mathematically correct and provides sharper learning rates for PCA and spectral support estimation. However, the clarity and organization of the paper has to be improved along with providing intuitive explanation of the theoretical results.
Author Feedback

Author rebuttal: * "I am wondering whether one needs the approach used in the paper to get sharp rates or one can still get such rates [...]

using classical Bernstein's inequality[...] without sophisticated tools like Theorem B.1"

While this approach would simplify the analysis, it produces looser bounds.
In particular, for p=2, alpha=1/2, and a decay rate r=2 (as in the example in the paper), it would be:

d_{1/2,2}(S_ρ,S_n) = O(n^{-1/4}) (using Theorem B.1)
d_{1/2,2}(S_ρ,S_n) = O(n^{-1/6}) (using Bernstein inequalities)



By using classical Bernstein inequalities in Hilbert spaces (e.g. Pinelis),
Lemma 7.2 would result in a range for t of q n^{-r/(r+1)} <= t <= ||C||,
implying k* = O(n^{1/(r+1)}) (instead of O(n^{1/r})), and thus Theorem 3.2 for k>=k* would become
d_{\alpha,p}(S_ρ,S^k_n) = O(n^{-alpha*r/(r+1) + 1/(p*(r+1))}) (compared with the sharper O(n^{-\alpha + 1/rp}) of Thm 3.2).

* "In theorem 3.2, corollary 3.3 and remark 3.4, it is not mentioned that \alpha > 1/(rp). Otherwise, the estimator wont be consistent."

This is an important observation that will be clarified in the text.
We note that this condition is not a restriction on the validity of the bound,
but is necessary for the subspace-learning problem to be feasible

(indeed, if \alpha <= 1/(rp) then |C^\alpha|_p = infty, and thus d_{\alpha,p}(S_ρ,S_n) = |(P - P^k_n)C^\alpha|_p = infty).

* "Can you please elaborate what is making the difference compared to [11], that the proposed formulation does not require a regularizer even from statistical point of view."

The work of [11] proves a U-shaped upper bound for d_{(r-1)/(2r),\infty}.
The bound's minimum converges to 0 (with n->\infty), but its value at k=n does not converge to 0.
Therefore, a form of regularization is needed to ensure consistency.
Our bound is instead L-shaped (no trade-off), and consistency is ensured for k=n whenever \alpha > 1/(rp).


* "Th. 3.1 in its current form seems to be for a fixed value of k[...], so perhaps a uniform bound on k would be more useful?"

As pointed out by the reviewer, one may in practice choose k based on the data, and a bound reflecting such a choice may be useful. We address this next.

* "[...]is it possible to give the error rate in terms of _observed_ covariance C_n and an empirical decay rate 'r_n'?"

The reviewer suggests an extension to the work in which the bound depends only on information that can be obtained from the sample data.
Our preliminary work in this direction indicates that it does seem to be possible to produce such a bound.

* One of the reviewers suggests moving some of the descriptions currently in section 4.1 earlier to sections 1 and 2, in order to further clarify the motivation